# Impact of COVID-19 by Pandemic Wave among Patients with Gastroenterology Symptoms in the Emergency Departments at a Medical Center in Taiwan

**DOI:** 10.3390/ijerph19127516

**Published:** 2022-06-20

**Authors:** Tony Kuo, Chun-Hao Liu, Cheng-Yu Chien, Chung-Cheng Yeh

**Affiliations:** 1Department of Hepatology and Gastroenterology, Chang Gung Memorial Hospital at Linkou, Taoyuan City 333, Taiwan; b9302028@cgmh.org.tw; 2College of Medicine, Chang Gung University, Taoyuan City 333, Taiwan; abucastor@hotmail.com; 3Department of Child & Adolescent Psychiatry, Chang Gung Memorial Hospital at Linkou, Taoyuan City 333, Taiwan; 4Graduate Institute of Management, Chang Gung University, Taoyuan City 333, Taiwan; rainccy217@gmail.com; 5Department of Emergency Medicine, Ton-Yen General Hospital, Zhubei City 302, Taiwan; 6Department of Emergency Medicine, Chang Gung Memorial Hospital at Keelung, Keelung City 204, Taiwan

**Keywords:** acute gastroenteritis, COVID-19, constipation, emergency department, gastro-intestinal bleeding

## Abstract

The COVID-19 pandemic has affected emergency department (ED) usage. This study examines changes in the number of ED visits for gastrointestinal (GI) bleeding and nonemergency GI conditions, such as acute gastroenteritis (AGE) and constipation, before the pandemic and at the peak and slack periods of the pandemic in Taiwan. This retrospective observational study was conducted at a referral medical center in northern Taiwan. We recorded the number of weekly ED visits for GI bleeding, AGE, and constipation from 2019 to 2021. We then compared the baseline period (calendar weeks 4–18 and 21–31, 2019) with two peak pandemic periods (period 1, calendar weeks 4–18, 2020; period 2, calendar weeks 21–31, 2021) and their corresponding slack periods. The decline in the number of ED visits during the two peak pandemic periods for GI bleeding (−18.4% and −30.2%) were not as substantial as for AGE (−64.1% and −76.7%) or for constipation (−44.4% and −63.6%), but GI bleeding cases were still significantly lower in number relative to the baseline. During the slack period, the number of ED visits for all three diagnoses rebounded but did not exceed the baseline. Our study revealed that there was a significant decline of GI complaint during the pandemic. This phenomenon was more prominent in nonemergency complaints (AGE and constipation) and less prominent in serious complaints (GI bleeding).

## 1. Introduction

The COVID-19 pandemic has affected patient healthcare seeking behavior, especially in the number of emergency department (ED) visits. In the United States, the total number of ED visits decreased by 42% during the beginning of the pandemic [1]. This decrease was most likely to be because patients feared getting COVID-19 and thus did not visit ED [2,3]. In an Italian study, Garrafa et al. revealed the ED visits decreased before lockdown (the “fear week”, after the news of the first COVID-19 contagion), which means the fear of contagion discouraged people accessing ED [4]. The decrease in ED visits number may potentially result in the delay of necessary medical intervention and even in increasing risk of mortality or morbidity. A study focused on obstetrical emergency revealed a significant delayed arrival to the emergency department during the pandemic, which led to the increase of urgent intervention [5]. Thus, knowing how the ED visits of each diagnosis was affected by different pandemic stages is important to patient’s welfare.

Taiwan has had two waves of the COVID-19 pandemic. The first wave was from 20 January 2020, when the first case of COVID-19 was discovered in Taiwan, to 7 June 2020, when local restrictions were eased. The second wave started from 20 April 2021, and then soon led to a national level 3 pandemic alert from 19 May to 26 July 2021. By contrast, life during slack periods barely differed from its pre-pandemic counterpart, setting Taiwan apart from most other countries. This feature made Taiwan a model to study how the pandemic changed people’s health care seeking behavior from baseline.

Many studies from different countries reported significant decline of ED visits during the pandemic [6,7,8,9]. Previous study of the pandemic has revealed significantly fewer recorded cases of some non-life-threatening conditions, such as exacerbation of congestive heart failure, but a nonsignificant change in the number of cases of life-threatening conditions, such as myocardial infarction [10]. In this study, we investigated whether this phenomenon also affected the number of ED visits for gastrointestinal (GI) complaints. GI complaints are common reasons for ED visits, with a wide variety in severity. Therefore, we hypothesize that the frequency of ED visits for serious GI complaints, such as GI bleeding, may be less affected by the pandemic compared with the frequency of nonemergency GI complaints, such as acute gastroenteritis (AGE) or constipation.

## 2. Materials and Methods

### 2.1. Study Design

This was a retrospective observational study conducted at a medical center, which is the largest hospital in northern Taiwan and was open for emergency referrals and walk-in patients during the pandemic. The study setting is a 3500-bed referral center with ED teams of 46 attending physicians. The study protocol was approved by the Institutional Review Board of Chang Gung Memorial Hospital (No. 202101591B0). Because our study did not collect any personal data, informed consent was waived by the review board. The study was conducted in compliance with the Declaration of Helsinki.

### 2.2. Definition of Study Periods

There were two study periods in, period 1 for calendar week 4–18 and period 2 for calendar week 21–31 (Figure 1). The first peak period (calendar weeks 4–18, 2020) was defined by the time of the discovery of the first COVID-19 case in Taiwan until 2 weeks before the second study period; the end point was defined as such to avoid overlap. The second peak period (calendar weeks 21–31, 2021) was defined by the period when the local level 3 pandemic alert was in effect. The baseline periods were defined as the corresponding calendar weeks in 2019 of the 2 peak periods (calendar weeks 4–18, 2019, for baseline period 1 and calendar weeks 21–31, 2019, for baseline period 2). The slack periods were defined by the corresponding calendar weeks during the pandemic without a local outbreak in Taiwan (calendar weeks 4–18, 2021, for slack period 1, and calendar weeks 21–31, 2020, for slack period 2).

### 2.3. ED Diagnoses

The total numbers of ED visits during the study periods were recorded. We retrieved the diagnoses data according to its ICD codes from the ED electronic medical report system. The ED diagnoses for GI bleeding (ICD-10 code: K922, diagnosed using clinical manifestation and laboratory data), AGE (ICD-10 code: K529, diagnosed using clinical manifestation), and constipation (ICD-10 code: 5900, diagnosed using clinical manifestation) were diagnosed by emergency physicians and separately cataloged for this study. Because this study focused on the reason why the patient visited the ED, instead of what happened during his admission, we only recorded ED diagnoses instead of final diagnoses. The ED diagnosis was also the basis for the clinical management in the ED and the payment from health insurance.

### 2.4. Statistical Analysis

Because of the relatively small sample sizes (*n* = 15 and 11 weeks), we used the median with the interquartile range (IQR) to present the number of ED visits for each week. ED visits during the peak period and the slack period were compared with their baseline counterparts using the Mann–Whitney test. We used the number of ED visits instead of the proportion of total ED visits because the ED visits during the peak pandemic period may have been affected by government policy (people needed to attend ED for COVID-19 PCR). Instead, we listed the total ED visit number and the degree declining from baseline as reference. We did not use multiple comparison in our study. Statistical significance was defined at a 2-tailed *p* < 0.05. Statistical analysis was performed using PASW statistics, version 18 (SPSS Inc., Chicago, IL, USA).

## 3. Results

The weekly ED visits of the three diagnoses are listed in Table 1 (period 1 week 4–18 in Table 1a, period 2 week 21–31 in Table 1b). Total number of ED visits was also listed as a reference.

### 3.1. Changes in the Number of ED Visits for GI Bleeding during the Pandemic

During calendar weeks 4 through 18, the median (IQR) numbers of cases were 49 (45–62), 40 (28–45) (*p* = 0.003), and 46 (43–50) (*p* = 0.081) in 2019 (baseline period), 2020 (peak period), and 2021 (slack period), respectively (Figure 2a). During calendar weeks 21 through 31, the median (IQR) numbers of cases were 43 (9–51), 43 (33–47) (*p* = 0.308), and 30 (29–32) (*p* = 0.001) in 2019 (baseline period), 2020 (slack period), and 2021 (peak period), respectively (Figure 2b).

### 3.2. Changes in the Number of ED Visits for AGE during the Pandemic

During calendar weeks 4 through 18, the median (IQR) numbers of cases were 103 (83–108), 37 (32–59) (*p* = 0.001), and 62 (43–73) (*p* < 0.001) in 2019 (baseline period), 2020 (peak period), and 2021 (slack period), respectively (Figure 2c). During calendar weeks 21 through 31, the median (IQR) numbers of cases were 90 (83–95), 50 (44–59) (*p* < 0.001), and 21 (20–28) (*p* < 0.001) in 2019 (baseline period), 2020 (slack period), and 2021 (peak period), respectively (Figure 2d).

### 3.3. Changes in the Number of ED Visits for Constipation during the Pandemic

During calendar weeks 4 through 18, the median (IQR) numbers of cases were 9 (8–15), 5 (3–7) (*p* < 0.001), and 8 (5–11) (*p* = 0.049) in 2019 (baseline period), 2020 (peak period), and 2021 (slack period), respectively. For calendar weeks 21 through 31, the median (IQR) numbers of cases were 11 (6–17), 11 (8–13) (*p* = 0.842), and 4 (1–5) (*p* < 0.001) in 2019 (baseline period), 2020 (slack period), and 2021 (peak period), respectively.

## 4. Discussion

We observed a decline in the number of ED visits for GI bleeding, AGE, and constipation during the COVID-19 pandemic. During the peak periods, fewer individuals visited the emergency department for all three conditions relative to the baseline; during the slack periods, fewer individuals visited the ED for GI bleeding, but not significantly so. This phenomenon was more prominent in nonemergency complaints and less prominent in serious complaints.

The change of ED visits pattern can be an indicator of the change in people’s health care seeking behavior [11]. Our study found a decrease of ED visits for the three GI diagnoses during the pandemic. Similar finding was reported by a retrospective review of a trauma center in the US, which demonstrated the ED visits for GI diagnoses had a 30.3% decline in April and a 26.7% decline in May of 2020, compared to baseline of 2019 [12]. There were many possible causes of the decline, such as lock-down of the city or limited health care capacity, but one possibility was the fear of being infected causing the delay in care seeking [11,13].

GI bleeding, a potentially lethal GI condition, may have been particularly concerning during the pandemic. However, fewer patients with GI bleeding were willing to seek help during the pandemic. A survey of 60 public hospitals reported substantially fewer (40.7%) cases of GI bleeding after a lockdown was implemented in Austria [14]. Another survey of 12 Chinese hospitals also revealed a 55.7% decline in the number of upper GI bleeding cases [15]. Besides their willingness to seek help, patients with GI bleeding may have further found that it was more difficult to access appropriate treatment during the pandemic. Although the mortality did not significantly change, patients with GI bleeding were less likely to receive endoscopy during the pandemic [16]. Performing endoscopy (including esophagogastroduodenoscopy and lower GI endoscopy), which is the conventional treatment for GI bleeding, became a problem during the pandemic because of the risk of being infected [17]. The combination of fear and restricted medical resources may have contributed to the delay in timely treatment for GI. A recent study revealed that patients with GI bleeding during the pandemic tended to have a more severe condition than in the pre-COVID era [18]. It is necessary to know how healthcare seeking behavior changed during the pandemic, and the cause and consequence of this phenomenon.

After the peak period of the pandemic, the public fear of COVID-19 reduced in the slack period. People with GI bleeding who delayed their treatment in the peak period began to seek medical care in the slack period. Our study reports significantly fewer ED visits for GI bleeding during the peak periods and a subsequent rebound during the slack periods. Hong Kong similarly experienced an initial decline in upper GI bleeding hospitalizations during the first wave of the pandemic followed by a rebound, according to one study [19]. That study also observed a negative correlation between COVID-19 cases and upper GI bleeding hospitalizations [19]. This decline–rebound trend was also reported in our previous study on cardiovascular ED visits [10], which indicated that the need for health care may have been postponed due to the fear of being infected.

Although SARS-CoV-2 can itself cause various GI symptoms [20], common GI diseases still occurred during the COVID-19 pandemic. Compared with serious diagnoses such as GI bleeding, nonserious complaints, such as those for AGE or constipation, were significantly less frequent during both the peak and slack periods of the pandemic. This finding was similar to our previous work on emergent and non-emergent patients with cardiovascular diagnoses in ED, which found urgent diagnoses (such as congestive heart failure), compared to emergent diagnoses (such as myocardial infarction), were more sensitive to the pandemic [10]. In UK, the percentage of gastroenteritis among all ED attendance dropped from 1.6% (pre-lockdown period) to 0.6% (lockdown period), while the percentage for appendicitis was relatively unaffected and only dropped from 0.5% to 0.3% [21]. This result indicated that the influence of COVID-19 pandemic was more pronounced in those with emergency diagnoses than non-emergency diagnoses.

However, there were other confounding factors affecting how people with GI complaints decided to visit ED or not. For example, the incidence of AGE and constipation may also have been affected during the pandemic because of lifestyle change such as less eating out or less exercise. This result may be affected by multiple factors and the causal inference was not direct. Honeyford et al. provided a conceptual framework to explain the changes in ED attendance, which includes fear as well as changes in access to other parts of the health care system, increased hand hygiene, and decreased physical activity [21]. Kostopoulou et al. also provided possible explanations for the decline of pediatric ED visit during the pandemic, which included (1) restriction of infection transmission, (2) avoidance of unnecessary ED visits, and (3) fear of exposure to COVID-19 [22]. In our study, we found a significant decline in AGE cases (approximate 40%) after the pandemic, even in the slack periods, which was more prominent than ED visits for constipation. This phenomenon may have resulted from the fear of COVID-19, but also from the reduction of the incidence of AGE. A large survey in the US also showed the number of AGE cases decreased by 52% in the first quarter of 2020 and remained at a low level for the remainder of the year [23]. Possible reasons for this phenomenon include the increased use of non-pharmacological interventions, such as face masks and hand washing, which also prevent the transmission of AGE. Further study on the change in prevalence of communicable diseases during the pandemic can provide more direct evidence of this observation.

This study has several limitations. First, the data were only from one medical center in Taiwan, so the result cannot be generalized to other reginal or local hospitals. Further multicenter surveillance or population-based studies are necessary to verify this preliminary finding. Second, the cross-sectional and descriptive research design of our study precluded causal inference. Our study design and material cannot give the actual reason why ED visits f GI bleeding decline, or if patients went to private clinics or just postponed their visit. A larger population-based longitudinal study is necessary in the future. Third, many confounding factors affecting the change in ED visits, such as disease prevalence or the restriction of out-patient service capacity in hospitals, were not evaluated in our study.

## 5. Conclusions

In conclusion, we found a significant decline in the number of ED visits for GI bleeding, AGE, and constipation during the pandemic. Although ED visits for GI bleeding were less frequent during the peak period, the decrease was not as significant as those for cases of AGE or constipation. Clinicians should be aware of the possible psychological influence of the pandemic on patients’ other health issues, besides COVID-19 itself.

## Figures and Tables

**Figure 1 ijerph-19-07516-f001:**
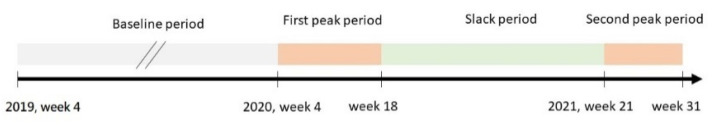
The definition of study periods. Period 1 week 4–18, and study period 2 week 21–31.

**Figure 2 ijerph-19-07516-f002:**
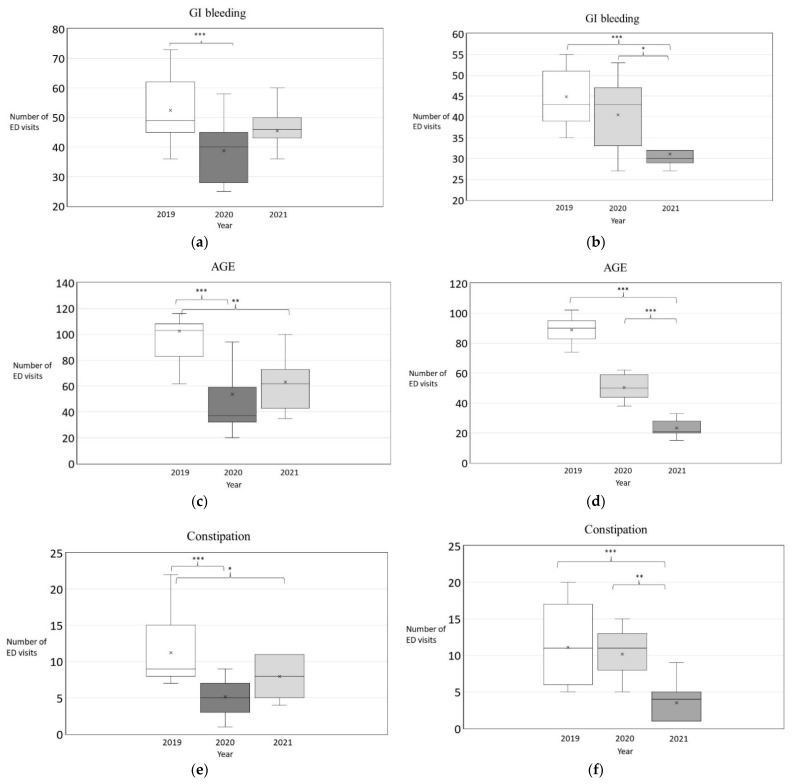
Boxplot of emergency department visits for (**a**) GI bleeding in period 1, week 4–18, (**b**) GI bleeding in period 2, week 21–31, (**c**) AGE in period 1, week 4–18, (**d**) AGE in period 2, week 21–31, (**e**) constipation in period 1, week 4–18, and (**f**) constipation in period 2, week 21–31. Note: Baseline period in white, peak pandemic period in dark gray, slack period in light gray. The x indicates the mean. * *p* < 0.05, ** *p* = 0.001, *** *p* < 0.001.

**Table 1 ijerph-19-07516-t001:** Number of ED visits per week, presented by median (IQR), for GI Bleeding, AGE, and Constipation during the pandemic in period 1 week 4–18 and period 2 week 21–31.

**a**								
**Period 1 of Week 4–18**								
	**Total ED Visits**		**GI Bleeding**		**AGE**		**Constipation**	
2019 (baseline)	3319 (3207, 3422)	reference	49 (45, 62)	reference	103 (83, 108)	reference	9 (8, 15)	reference
2020 (peak)	2007 (1794, 2455) −39.5%	*p* < 0.001	40 (28, 45) −18.4%	***p* = 0.003**	37 (32, 59) −64.1%	***p* = 0.001**	5 (3, 7) −44.4%	***p* < 0.001**
2021 (slack)	2573 (2484, 2700) −22.5%	*p* < 0.001	46 (43, 50) −6.1%	*p* = 0.081	62 (43, 73) −39.8%	***p* < 0.001**	8 (5, 11) −11.1%	***p* = 0.049**
**b**								
**Period 2 of Week 21–31**								
	**Total ED Visits**		**GI Bleeding**		**AGE**		**Constipation**	
2019 (baseline)	3465 (3292, 3567)	reference	43 (39, 51)	reference	90 (83, 95)	reference	11 (6, 17)	reference
2020 (slack)	2582 (2448, 2658) −25.5%	*p* < 0.001	43 (33, 47) 0%	*p* = 0.308	50 (44, 59) −44.4%	***p* < 0.001**	11 (8, 13) 0%	*p* = 0.842
2021 (peak)	2543 (2211, 2780) −26.6%	*p* < 0.001	30 (29, 32) −30.2%	***p* = 0.001**	21 (20, 28) −76.7%	***p* < 0.001**	4 (1, 5) −63.6%	***p* < 0.001**

Note: AGE = acute gastroenteritis, ED = emergency department, GI bleeding = gastrointestinal bleeding.

## Data Availability

Not applicable.

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
