# Peer review of "Impact of COVID-19 by Pandemic Wave among Patients with Gastroenterology Symptoms in the Emergency Departments at a Medical Center in Taiwan"

_ijerph, 2022, doi:10.3390/ijerph19127516_

Round 1

Reviewer 1 Report

The authors found a significant decline in the number of ED visits for GI bleeding, AGE, and constipation during the pandemic. However, I can not find any result to prove this sentence "This may indicate an unmet gap between patient needs and access to treatment during the pandemic."  in Conclusion.
Besides, it seems that the authors wanted to discuss the change in patient health-seeking behavior. But their finding did not provide any suggestions to physicians or medical managers. 

Author Response

Q1

The authors found a significant decline in the number of ED visits for GI bleeding, AGE, and constipation during the pandemic. However, I can not find any result to prove this sentence "This may indicate an unmet gap between patient needs and access to treatment during the pandemic."  in Conclusion.

Thank you for pointing this out. To avoid overstatement, we revised our conclusion section as below.

“Clinicians should be aware of the possible psychological influence of the pandemic on patients’ other health issues, beside of COVID-19 itself.”

Q2

Besides, it seems that the authors wanted to discuss the change in patient health-seeking behavior. But their finding did not provide any suggestions to physicians or medical managers. 

We revised our statement in the Discussion section and the Conclusion section, and we also cited a new reference to support our statement.

In the Discussion section,

“The change of ED visits pattern can be an indicator of the change of people’s health care seeking behavior.”

In the Conclusion section,

“Clinicians should be aware of the possible psychological influence of the pandemic on patients’ other health issues, beside of COVID-19 itself.”

Reference:

Nab M, van Vehmendahl R, Somers I, Schoon Y, Hesselink G. Delayed emergency healthcare seeking behaviour by Dutch emergency department visitors during the first COVID-19 wave: a mixed methods retrospective observational study. BMC Emerg Med. 2021;21(1):56.

Reviewer 2 Report

Table: it is a little confusing to have Table 1a ordered peak then slack and Table 1b ordered slack then peak. Consider reformatting the table(s) to make it more clear

Line 164: why? support this statement

Lines 166-169: Are there any reports to indicate that failure to present in other countries resulted in worse outcomes? Correlation with mortality data with cause of deaths associated with GI bleeding? 

Line 180: overstating - it is not clear that these are the same "people" though it is within the same population; reword

Line 194: acute stroke is considered an emergent diagnosis on par with myocardial infarction in terms of the necessity of timeliness of treatment - please reassess this

Line 199: is this the opposite of what you intend? that the impact was actually greater on non-emergent conditions than emergent diagnoses? Above you state that AGE attendance dropped 1% and that appendicitis dropped less

Line 223: clarify; the differentiation between ED diagnosis and final diagnosis is not clear to me - can you provide background on this process and who makes the final diagnosis and under what circumstances? is there a difference between discharged and hospitalized patients for this?

Overall, this study investigates an interesting phenomenon that has been observed in a variety of conditions surrounding covid-19, regarding ED presentation variability for different disease states. This topic is being explored in varying degrees of granularity, but generally widely studied at this time. The authors focus on GI complaints. Generally this paper would benefit from improved clarity on some of the definitions as noted above.  Additionally, I am curious about the decision to report the comparisons of ED visit counts rather than as a proportion of overall visits. I think that there may be some benefit in exploring both.  For example, ED visits were still down by 1000 at a time when there was minimal change in the count of ED visits for GI bleed, yet the difference was not identified as significant. I think it at least warrants some additional discussion on the proportion of presentations. 

Author Response

Q1

Table: it is a little confusing to have Table 1a ordered peak then slack and Table 1b ordered slack then peak. Consider reformatting the table(s) to make it more clear

Thank you for your suggestion. Indeed, baseline-peak-slack and baseline-slack-peak order was a little bit confusing. However, we arrange our table in this order by temporality, means 2019-2020-2021. In period 1, week 4-18 (table 1a), 2020 was the peak and 2021 was the slack period, but it was reversed in period 2 (week 21-31, table 1b), while 2020 was the slack and 2021 was the peak. This order was according to Taiwanese pandemic timeline, so we felt sorry for make it confusing.

Q2

Line 164: why? support this statement

We revised our wording as the following to avoid over-stating.

“There were many possible causes of the decline, such as lock-down of the city or limited health care capacity, but one possibility was the fear of being infected causing the delay of care seeking”

Reference

1.     Nab M, van Vehmendahl R, Somers I, Schoon Y, Hesselink G. Delayed emergency healthcare seeking behaviour by Dutch emergency department visitors during the first COVID-19 wave: a mixed methods retrospective observational study. BMC Emerg Med. 2021;21(1):56.

2.     Lai AY, Sit SM, Wu SY, et al. Associations of Delay in Doctor Consultation With COVID-19 Related Fear, Attention to Information, and Fact-Checking [published correction appears in Front Public Health. 2022 Feb 10;10:847603]. Front Public Health. 2021;9:797814.

Q3

Lines 166-169: Are there any reports to indicate that failure to present in other countries resulted in worse outcomes? Correlation with mortality data with cause of deaths associated with GI bleeding? 

To our knowledge, there was no study demonstrated the direct causal inference on the increased mortality of GI bleeding and the failure to present to medical care during the pandemic. We cited a new reference about patients with GI bleeding were less likely to receive endoscopy during the pandemic, but the mortality was no significantly changed. We added this description as below.

“Although the mortality was not significantly changed, patients with GI bleeding were less likely to receive endoscopy during the pandemic.”

Reference.

Khan R, Saha S, Gimpaya N, et al. Outcomes for upper gastrointestinal bleeding during the first wave of the COVID-19 pandemic in the Toronto area. J Gastroenterol Hepatol. 2022;37(5):878-882.

Q4

Line 180: overstating - it is not clear that these are the same "people" though it is within the same population; reword

Thank you for pointing out. We changed our wording as the following.

“It is necessary to know how health care seeking behavior changed during the pandemic, and the cause and consequence of this phenomenon.”

Q5

Line 194: acute stroke is considered an emergent diagnosis on par with myocardial infarction in terms of the necessity of timeliness of treatment - please reassess this

Indeed, timeliness of treatment is critical in both acute stroke and MI, e.g. “time is muscle” and “time is brain”. Both acute stroke and MI were saw emergency “in-hospital”. However, “pre-hospital” delay was more common in acute stroke than MI [1]. The cited study [2] focused on people’s behavior instead of medical professionals’ behavior, so “pre-hospital delay” was more interesting than “in-hospital” delay. Moreover, symptoms of MI include pain, but symptoms of acute stroke did not and sometimes were gradual and less severe. That made people ignored the severity of acute stroke. All the above may result in the significant decline of ED visits due to acute stroke during the pandemic.

However, to avoid confusing and misleading, we deleted the “acute stroke” in the sentence.

1.     Herlitz J, Wireklintsundström B, Bång A, Berglund A, Svensson L, Blomstrand C. Early identification and delay to treatment in myocardial infarction and stroke: differences and similarities. Scand J Trauma Resusc Emerg Med. 2010;18:48.

2.     Yeh C.C., Chien C.Y., Lee T.Y., Liu C.H. Effect of the COVID-19 pandemic on emergency department visits of patients with an emergent or urgent diagnosis. Int J Gen Med. 2022;15:4657-4664. doi:10.2147/IJGM.S362615.

Q6

Line 199: is this the opposite of what you intend? that the impact was actually greater on non-emergent conditions than emergent diagnoses? Above you state that AGE attendance dropped 1% and that appendicitis dropped less

Our hypothesis was the pandemic affect non-emergency diagnoses more than emergency diagnoses. This reference also demonstrated similar phenomenon, that AGE dropped more than appendicitis did.

Q7

Line 223: clarify; the differentiation between ED diagnosis and final diagnosis is not clear to me - can you provide background on this process and who makes the final diagnosis and under what circumstances? is there a difference between discharged and hospitalized patients for this?

If the patient was admitted to ICU or ward, his or her final diagnosis may be different from its initial presentation in ED, for example, a patient initially presented to ED due to GI bleeding, but then complicated with AMI during his stay in hospital. However, to prevent misleading, we delete the description about final diagnosis from the “limitation” section, and describe it in the “method” section as below.

“Because this study focused on the reason why the patient visited the ED, instead of what happened during his admission, we only recorded ED diagnoses instead of final diagnoses. The ED diagnosis was also the basis for the clinical management in the ED and the payment from health insurance.”

Q8

Overall, this study investigates an interesting phenomenon that has been observed in a variety of conditions surrounding covid-19, regarding ED presentation variability for different disease states. This topic is being explored in varying degrees of granularity, but generally widely studied at this time. The authors focus on GI complaints. Generally this paper would benefit from improved clarity on some of the definitions as noted above.  Additionally, I am curious about the decision to report the comparisons of ED visit counts rather than as a proportion of overall visits. I think that there may be some benefit in exploring both.  For example, ED visits were still down by 1000 at a time when there was minimal change in the count of ED visits for GI bleed, yet the difference was not identified as significant. I think it at least warrants some additional discussion on the proportion of presentations. 

Thank you for your comment. We used “the number of ED visits of a specific diagnosis” instead of “the proportion of overall ED visit” because of some reasons listing below:

(1)   By government’s policy, people need to attend ED for COVID-19 PCR. Thus, during the peak period, there were many ED visits only for COVID-19 PCR instead of a specific complaint. It was different from baseline and slack period, while people went to ED mostly because of physical discomfort.

(2)   As above reason, we do not use “proportion of overall ED visit” in our study, but we also listed the total ED visit number in Table 1 as a reference.

(3)   We also list the degree of declining (in %) compared to baseline period. For example, while the overall ED visit decreased 40% but the ED visits for GI bleeding only decreased 20%, we can interface that the impact of pandemic on GI bleeding was less than overall ED visit.

(4)   We followed the methodology used by previous studies [1-3], while most of the previous studies interested in this topic used ED visit number instead of the proportion.

We had summarized the reasons in our method section as the following.

“We use the number of ED visits instead of the proportion of total ED visits because the ED visits during peak pandemic period may affected by government’s policy (people need to attend ED for COVID-19 PCR). Instead, we listed total ED visit number and the degree of declining from baseline as reference.”

Reference:

1.     Oikonomou E, Aznaouridis K, Barbetseas J, et al. Hospital attendance and admission trends for cardiac diseases during the COVID-19 outbreak and lockdown in Greece. Public Health. 2020;187:115-119.

2.     Schmiderer A, Schwaighofer H, Niederreiter L, et al. Decline in acute upper gastrointestinal bleeding during COVID-19 pandemic after initiation of lockdown in Austria. Endoscopy. 2020;52(11):1036-1038.

3.     Schwarz V, Mahfoud F, Lauder L, et al. Decline of emergency admissions for cardiovascular and cerebrovascular events after the outbreak of COVID-19. Clin Res Cardiol. 2020;109(12):1500-1506.

Reviewer 3 Report

In general, the paper by Kuo et al concentrates on important topic regarding pandemic situation and giving a perspective of its influence on visits and needs for Emergency Departments. Overall, the manuscript is well written and methods clearly outlined. However, I suggest to write results in the present form, it would help to read better.

I have few minor comments:

1.       It would help to understand the aim if you specify the reason why GI conditions were chosen. Just shortly before the sentence on line 56.

2.       On line 97, statistical analysis – please clarify what means “(n = 15 and 11 weeks)” – is it number of patients or weeks or something else? Can’t see those numbers in the Tables.

3.       Format the table so that you won’t be using 1a and 1b – for example leave the top heading and then add the cell for “period 1 of week 4-18”, then what you already have, then again extra cell with “period 2 of week 21-31” and then what you already have. It should make it easier to read.

4.       Are the statistics adjusted to multiple comparison? If not, please state it in the methods.

5.       The text of the results should be simplified to have a clearer message. In my opinion, not all numbers should be repeated (they are already in the tables), only most relevant ones.

6.       Very minor comment figures’ axes names should be in the middle.

7.       Table one says weekly visits, on the figure number of visits. For me it seems that they are the same. Please recheck which one they are and correct accordingly.

8.       I couldn’t find the stats for total ED visits. It would be helpful to have them.

9.       In discussion, the sentences in the first paragraph containing % should be rephrased because at the moment it sounds like the reduction was from 44.4 to 76.7% etc and it’s confusing.

10.   In the discussion – please discuss if there is less GI bleeding cases in ED then what is happening with those people: are they ending up family physicians or … What have other studies shown?

Author Response

Q1

In general, the paper by Kuo et al concentrates on important topic regarding pandemic situation and giving a perspective of its influence on visits and needs for Emergency Departments. Overall, the manuscript is well written and methods clearly outlined. However, I suggest to write results in the present form, it would help to read better.

I have few minor comments:

1.       It would help to understand the aim if you specify the reason why GI conditions were chosen. Just shortly before the sentence on line 56.

Thank you for your comment. We added a statement to clarify the reason we focused on GI complaints. First, GI complaints are common reason for ED visits. Second, GI complaints have a large variety of severity, from AGE or constipation (non-emergency) to GI bleeding (emergency). We added the statement as below.

“GI complaints are common reasons for ED visits and have a large variety of its severity.”

Q2

2.       On line 97, statistical analysis – please clarify what means “(n = 15 and 11 weeks)” – is it number of patients or weeks or something else? Can’t see those numbers in the Tables.

The number (n) was how many weeks in each period. Because of the epidemic periods were quite short in Taiwan (3-4 months) then back to slack period, the n was relatively small for t-test.

Q3

3.       Format the table so that you won’t be using 1a and 1b – for example leave the top heading and then add the cell for “period 1 of week 4-18”, then what you already have, then again extra cell with “period 2 of week 21-31” and then what you already have. It should make it easier to read.

We had revised our table as your suggestion.

Q4

4.       Are the statistics adjusted to multiple comparison? If not, please state it in the methods.

We state that “We did not use multiple comparison in our study” in our method section.

Q5

5.       The text of the results should be simplified to have a clearer message. In my opinion, not all numbers should be repeated (they are already in the tables), only most relevant ones.

We left number of ED visits with IQR and p-value in our result section. Percentages were deleted to make it clearer.

Q6

6.       Very minor comment figures’ axes names should be in the middle.

We have revised our figure.

Q7

7.       Table one says weekly visits, on the figure number of visits. For me it seems that they are the same. Please recheck which one they are and correct accordingly.

Yes, they mean the same thing. We revised the wording of Table 1.

Q8

8.       I couldn’t find the stats for total ED visits. It would be helpful to have them.

We had added it in our table 1.

Q9

9.       In discussion, the sentences in the first paragraph containing % should be rephrased because at the moment it sounds like the reduction was from 44.4 to 76.7% etc and it’s confusing.

Thank you for your comments. We delete the unnecessary sentences in the discussion section which were misleading and already shown in the result section.

Q10

10.   In the discussion – please discuss if there is less GI bleeding cases in ED then what is happening with those people: are they ending up family physicians or … What have other studies shown?

Thank you for your suggestion. We tried our best to find literature but there was no study can answer this question. A national-wide population-based study design is necessary to now the dynamic between health care facilities. We had listed this in our limitation section as below.

“Second, the cross-sectional and descriptive research design of our study precluded causal inference. Our study design and material also cannot tell the actual reason why ED visit of GI bleeding decline, or if they went to private clinics or just postponed their need. A larger population-based longitudinal study is necessary in the future..”

Round 2

Reviewer 1 Report

The authors have revised their statement in the Discussion and Conclusion sections and cited a new reference to support the idea. I have no further comment.

Reviewer 2 Report

I appreciate the work to address the comments from the original version, particularly the updates to the tables.

Minor grammatical errors to be fixed.

This manuscript is a resubmission of an earlier submission. The following is a list of the peer review reports and author responses from that submission.